

# Effects of terracing on soil water and canopy transpiration of Chinese pine plantation in the Loess Plateau, China

Handan Zhang[1,2], Wei Wei[1*], Liding Chen[1], Lixin Wang[3]

[1] State Key Laboratory of Urban and Regional Ecology, Research Center for

Eco-Environmental Sciences, Chinese Academy of Sciences, Beijing 100085, China.

[2] University of Chinese Academy of Sciences, Beijing 100049, China.

[3] Department of Earth Sciences, Indiana University-Purdue University Indianapolis

(IUPUI), Indianapolis 46202, United States

*Corresponding author: Wei Wei (weiwei@rcees.ac.cn), Tel.: +86-10-6291-8673,

Fax: +86-10-6291-3840.



**Abstract**
Terracing has long been considered one of the most effective measures for soil water
conservation and site improvement. However, the quantitative effects of terracing on soil water
dynamics and vegetation water use have not been reported. To fill these knowledge gaps, in this
study, soil water content and canopy transpiration were monitored in both terrace and slope
environments in the semiarid Loess Plateau of China in 2014 and 2015. Results showed that
terracing increased soil water content of different soil layers. Mean soil water content of the
terrace site was 25.4% and 13.7% higher than that in the slope site in 2014 and 2015, and canopy
transpiration at the terrace site increased by 9.1% and 4.8%, respectively. Canopy conductance at
the terrace site was 3.9% higher than that at the slope site and it decreased logarithmically with
vapor pressure deficit. This study highlighted the critical role of terracing in increasing the soil
water content and mitigating water stress in semiarid environments. Thus, terracing has the
potential to enhance sustainable vegetation restoration in water-limited regions.
**Keywords**: terracing; sap flux density; canopy conductance; water stress; Loess Plateau
**1 Introduction**
Terraces constitute a crucial engineering measure to control erosion, raise crop yields, and
maintain sustainable agroforestry. By leveling hillslopes, terraces seek to create better planting
surfaces for mitigating water loss and conserving soil (LaFevor, 2014; Zhang et al., 2014).
Terracing has been established as the main measure for soil and water conservation for fields
with gradients under 25 degrees (Li et al., 2011; Li et al., 2012b). It has been determined that
terracing in such locations can reduce both flood runoff and the sediment transport modulus (Bai
et al., 2015; Li et al., 2014a), and that the soil water conditions can be improved noticeably
(Courtwright and Findlay, 2011; Huo and Zhu, 2013).



Transpiration as an important role part of the soil-plant-atmosphere continuum (Newman et al.,
2010) has considerable implications regarding forest management and water yields (Bosch et al.,
2014; Brito et al., 2015; Chang et al., 2014b), especially in regions where transpiration is a
fundamental datum for understanding the ecophysiology of planted forests (Wang et al., 2012a).
It is also central to the construction of an ecosystem-level water balance (Yang et al., 2009). Sap
flow measurement can provide insights on environmental limitations and it yields results
comparable with the estimates of water use for entire forest ecosystems (Chen et al., 2014b;
Chirino et al., 2011; Du et al., 2011; Kim et al., 2014). Previous studies have shown that sap flow
characteristics vary with species and growth status, as well as with meteorological,
environmental, and edaphic features (Brito et al., 2015; Du et al., 2011). In areas with
insufficient water, soil water conditions can restrict many physiological processes (Li et al.,
2014b). Plants in these areas tend to deepen and extend their root systems to exploit substantial
quantities of soil water for transpiration (Chen et al., 2014c; Limousin et al., 2009). Stomatal
closure as an important physiological process was employed by plants to regulate water use and
to prevent their hydraulic system from irreversible damage (Chirino et al., 2011). Sap flow
reduction caused by stomatal closure is considered to be the preliminary response of canopy
transpiration to water stress. Under water-sufficient conditions, differences in vapor pressure
deficit (VPD) determine the transpiration amount (Chen et al., 2014b). However, transpiration is
restricted by the plant's hydraulic conductance capacity and cannot exceed the amount of water
that can be obtained from the soil. Soil water influences stand transpiration through the water
fluxes within the root zone and the percolation of soil profile caused by different rainfall regimes
(Chen et al., 2014c). Based on pot experiments, Cui (2012) concluded that sap flow rates
dropped 84.7% under severe water stress (5.33%) compared with that under non-stress (19.78%)



conditions. Under saturated conditions, sap flow rates were found to reach 10 times of those in
the dry season (Nie et al., 2005).
The semiarid Loess Plateau region of China has experienced long-term serious soil erosion,
vegetation degradation, and water loss (Zhang et al., 2008). Intense soil erosion has resulted in
the decline of land productivity (under traditional agriculture) and environmental degradation
(Wang et al., 2010). Because of the depletion of soil moisture and water shortages, there are
many "dwarf and aged" trees in this region (Li et al., 2013). Hence, with the objectives of
controlling erosion and conserving water resources, many investigations have been conducted
into a wide range of soil management practices, including structural, agronomic, and biological
measures (Jin et al., 2014; Yuan et al., 2016). Among these, terraces are a well-developed
structural practice. Unlike native plants, many introduced species of vegetation usually have
higher water demands (Chen et al., 2008; Yang et al., 2009). Thus, local soils have become
extremely dry in both deep and shallow layers, diminishing the expected positive effects of
afforestation in controlling soil erosion and improving the regional environment (Wang et al.,
2012b; Yang et al., 2012). By analyzing four introduced plant species (*Pinus tabulaeformis*,
*Robinia pseudoacacia*, *Caragana korshinskill* and *Hippophae rhamnoides*), Jian et al. (2015)
drew the conclusion that in semiarid loess hilly areas, precipitation cannot meet the water loss
caused by evapotranspiration in slope-scale. However, few studies have considered the effects of
terracing on plant growth, nor its implications for regional ecological restoration.
This paired-site study focused on a small catchment in the western Loess Plateau of China to
examine the effects of terracing on the soil water content and canopy transpiration. Similarly
aged specimens of Chinese pine (*P. tabulaeformis*), being one of the main artificial plants in the
area, were planted in both terrace and slope plots. The specific aims of this study were to (1)





examine the effects of terracing on soil moisture dynamics; (2) identify the effect of terracing on
canopy transpiration.
**2 Materials and methods**
**2.1 Site description**
The study area was located in Anjiapo catchment in Dingxi County of Gansu Province, in the
western part of the Loess Plateau in China (35°33′–35°35′N, 104°38′–104°41′E). This region has
a continental arid temperate climate with mean annual temperature and mean annual rainfall of
6.3 ℃ and 421 mm, respectively (1956–2010 period). Most of the rain falls during the summer
months in the form of thunderstorms. The mean annual pan evaporation reaches 1515 mm. The
soil type belongs to calcic Cambisol (FAO, 1990), developed from loess material, with the
average soil depth varying from 40 to 60 m. In this area, deep percolation can be neglected and
groundwater is unavailable for vegetation growth and restoration. Therefore, rainfall is the only
water source available for plants. The predominant vegetation types in the study area are native
grasses and introduced plants. In this study, two adjacent stands were chosen for the experiment:
one with natural sloping topography and the other that has been terraced for over 30 years (Fig.
1). Both sites were planted with specimens of *P. tabulaeformis*, a planted tree species typical of
the region (Chen et al., 2010; Wei et al., 2015).
**2.2 Environmental observation**
Micrometeorological data such as air temperature ($T$, ℃), solar radiation ($R_a$, W · m$^{-2}$), relative
humidity ($RH$, %), and precipitation ($P$, mm) were obtained using a Vantage Pro2 automatic
weather station (Davis Company, USA) located in an open space about 500 m from the site.
Vapor pressure deficit (VPD, kPa) was calculated based on the air temperature and relative
humidity as:



$$VPD = 0.611 \times \exp(\frac{17.27T}{237.3+T})(1-\mathrm{RH})$$
(1)

Soil water content was monitored continuously using a HOBO U30 (Onset Computer
Corporation, Bourne, USA) from 2014 to 2015 within the upper 100 cm of the soil profile. There
were five probes in each instrument set to depths of 10, 30, 50, 70, and 90 cm, respectively.
Relative extractable water (REW) was calculated as:
$$REW = (\theta - \theta_{\min}) / (\theta_{\max} - \theta_{\min})$$
(2)

where $\theta_{\max}$ and $\theta_{\min}$ are the maximum and minimum soil water content, respectively. The
value of REW varies between 0 and 1. Following Bréda et al. (2006), soil water conditions were
classified into severely stressed (REW = [0, 0.1]), moderately stressed (REW= [0.1, 0.4]), and
non-stressed (REW = [0.4, 1]).
**2.3 Sap flux and transpiration measurements**
Sap flux was monitored continuously from June 5, 2014 to October 10, 2015. At each studied
site, six individuals of *P. tabulaeformis* with different diameters at breast height (DBH, cm) were
selected, which represent the size classes within the site (Table 1). Sap flow was measured with
the improved Granier's thermal dissipation probe technique (Granier, 1985). The detailed
procedure for measuring sap flow was described in Zhang et al. (2015).
Sap flux density ( $SF_d$ ) was calculated by an empirical calibration equation:
$$SF_d = 0.714 \times \left( \frac{\Delta T_{\max} - \Delta T - (\Delta T_{R1} + \Delta T_{R2})/2}{\Delta T - (\Delta T_{R1} + \Delta T_{R2})/2} \right)^{1.231}$$
(3)

where $SF_d$ is sap flux density (mL cm$^{-2}$ min$^{-1}$); $\Delta T$, $\Delta T_{R1}$, and $\Delta T_{R2}$ each represent the
temperature difference between probes (Zhang et al., 2015); and $\Delta T_{\max}$ is the maximum value



of $\Delta T$ in cases when the tree was saturated, i.e., no radial tree-trunk increment, air humidity of
100%, and transpiration near zero.
Sap flux ($SF$, kg day$^{-1}$) was obtained by the multiplication of sap flux density and sapwood area
($A_s$, cm$^2$), neglecting the differences in radial profile (Chang et al., 2014a). It was calculated as:

117                             $$SF = 1.44 SF_{d} A_{s} \qquad (4)$$

Canopy transpiration ($E_c$, mm day$^{-1}$) was obtained from the $SF$ and crown projected area ($A_c$, m$^2$)
(Chang et al., 2014a) as:

120                             $$E_{c} = SF/A_{c} \qquad (5)$$

Mean daily canopy conductance ($g_c$, mm s$^{-1}$) was estimated from canopy transpiration ($E_c$,
mm h$^{-1}$) by using a simplified inverted Penman-Monteith equation (Luis et al., 2005):

123                             $$g_{c} = \gamma \lambda E_{c}/\rho c_{p} VPD \qquad (6)$$

where $\gamma$ is the psychometric constant (kPa $\cdot °C^{-1}$), $\lambda$ is the latent heat for vaporizing (MJ kg$^{-1}$),
$\rho$ is the air density (kg m$^{-3}$), $c_p$ is the specific heat capacity of air (MJ kg$^{-1} \cdot °C^{-1}$), and the
VPD (kPa) is calculated from the air temperature and relative humidity. The value of $g_c$ was
assumed as approximate average stomatal conductance and considered to reflect the
physiological control of tree transpiration.
**2.4 Statistical analysis**
DBH, sapwood area, and crown projected area were compared using the student *t* test. For the
comparison of soil water content and canopy transpiration dynamics, non-parametric tests of
significance were used because of the autocorrelations in the time series data. The Wilcoxon
rank sum test, also known as the Mann-Whitney *U* test, was used to test the differences in soil
water content and canopy transpiration between the terrace and slope sites. Curve fitting was





performed using the OriginPro Version 8.0 software (OriginLab Corporation, USA) to establish
the relationship between canopy transpiration and soil water content, and between canopy
conductance and VPD. Statistical analyses were run using the SPSS version 17.0 software (SPSS
Inc., Chicago, IL, USA), for which, the significance level was set at 0.05.
**3 Results**
**3.1 Soil water content**
Under the same climatic conditions, soil water content showed differences between the natural
slope and terraces (Fig. 2). Data from the shallow layer (0–20 cm) were not analyzed because of
anthropogenic disturbance. In both years, statistically significant ($p < 0.05$) higher soil water
content was observed at the terrace site than that at the slope site (Fig. 2 a and b).
Depth-averaged soil water content of the terrace site was approximately 25.4% and 13.7% higher
than that at the slope site in 2014 and 2015, respectively. Moreover, the mean soil water contents
at both sites were higher in 2015 than that in 2014. Temporal variations of REW between 20–
100 cm (Fig. 2 c and d) indicated that soil water conditions were stressed (REW < 0.4) in both
sites during the two consecutive growing seasons. However, REW was 113.1% more at the
terrace site compared with that at the slope site during the two years. It was noted that soil water
was severely stressed (REW < 0.1) in the slope site, whereas terracing improved the conditions
significantly.
**3.2 Canopy transpiration**
The diurnal variations of sap flux density ($SF_d$) are shown in Fig. 3. In the growing season, *P.*
*tabulaeformis* had similar trends of variation at both sites, i.e., high flux density in the daytime
and low flux density at night. It varied between 0.02 and 0.23 mL cm$^{-2}$ min$^{-1}$ at the terrace site
and between 0.02 and 0.18 mL cm$^{-2}$ min$^{-1}$ at the slope site. *P. tabulaeformis* had 20.2% higher





maximum sap flux density at the terrace site compared with that at the slope site. Canopy
transpiration was found to be 9.1% and 4.8% higher ($p < 0.05$) at the terrace site than that at the
slope site in 2014 and 2015, respectively. Annual variation analysis showed that the cumulative
canopy transpiration at both sites was higher in 2014 than that in 2015 (Fig. 4). In the naturally
sloping site, the cumulative canopy transpiration was 138.6 mm (32.9% of potential
evapotranspiration (PET)) in 2014 and 107.6 mm (24.9% of PET) in 2015. The corresponding
proportions at the terrace site were 35.7% and 26.0% in 2014 and 2015, respectively. Variation
in canopy transpiration between the slope and terrace sites increased with soil water content
variation ($p < 0.0001$, $R^2 = 0.20$; Fig. 5).

### 3.3 Canopy conductance

We classified canopy conductance into two levels based on soil water conditions: REW > 0.1
(Fig. 6 a and b) and REW < 0.1 (Fig. 6 c and d). The relationships between canopy conductance
and solar radiation, between canopy conductance and VPD under corresponding soil water
conditions are shown in Figure 6. It exhibits that canopy conductance declined logarithmically
with VPD (Fig. 6 b and d), and there is no significant relationship between canopy conductance
and solar radiation (Fig. 6 a and c). When soil water conditions changed from wet (REW > 0.1)
to dry (REW < 0.1), canopy conductance reduced by 12.3% and 24.7% at the slope and terrace
sites, respectively. Meanwhile, canopy conductance of *P. tabulaeformis* at the terrace site was up
to 3.9% higher than that at the slope site. The frequency of the $SF_d$ peak time suggested that *P.
tabulaeformis* suppressed $SF_d$ under high VPD conditions at both slope and terrace sites (Fig. 7).
The maximum $SF_d$ ($SF_{d, max}$) was relatively similarly distributed before 14:00 local time (LT), i.e.,
61.1% at the slope site and 59.2% at the terrace site. However, around 16:00 LT, closer to the
most frequent peak time of VPD, the proportion of $SF_{d, max}$ at the slope site was 33.3% less than





that at the terrace site. Therefore, under the same conditions, terracing was found to alleviate the
sensitivity of stomatal response to ambient air humidity.
**4 Discussion**
**4.1 Effects of terracing on soil water recharge**
A statistically significant ($p < 0.05$) higher soil water content was found at the terrace site
compared with that at the slope site (Fig. 2). Terraces, which interrupt natural slopes with a
series of gentle benches, can decrease the connectivity and integrity of overland flow, prolong
the residence time of water, and increase the infiltration (Molina et al., 2014). According to
Zhang et al. (2005), the soil profile in a terrace can be divided into three layers: the fast changing
layer, activity layer, and relatively stable layer. Water storage in the fast changing layer of a
terrace can be 7.2% higher than that in sloping land (Huo and Zhu, 2013). Similar to Wang et al.
(2014a), soil water content of the terrace site in this study was significantly higher ($p < 0.05$)
than that at the slope site within 100 cm in each layer. The depth-averaged soil water content in
the terrace site was up to 25.4% higher than that at the slope site (Fig. 2). Similar results have
been obtained in studies that compared the effects of contour bench terrace systems in the
semiarid Negev in Israel (Stavi et al., 2015), examined terrace characteristics (Engdawork and
Bork, 2014), and detected the impact of restoring degraded terraces (LaFevor, 2014). Previous
works have reported that approximately 20% (to a potential 200%) of total surface rainwater
could infiltrate into underground soil layers after terracing (Courtwright and Findlay, 2011), and
that 1.13 times more rainfall can be stored in a terraced system than that in sloping land (Li et al.,
2012a). The low REW indicated that the study area is under severe water stress, whereas the
large difference between the two sites (113.1 % more REW in the terrace site) suggested that the
construction of terraces could help increase soil water content.





**4.2 Effects of terracing on canopy transpiration**
According to the results, the maximum sap flux density of *P. tabulaeformis* at the terrace site
was 20.2% higher than that at the slope site under the same climatic conditions (Fig. 3). During
the growing seasons, mean daily canopy transpiration was up to 9.1% ($p < 0.05$) higher at the
terrace site than that at the slope site (Fig. 4). Similarly, Pataki et al. (2000) found an observed
decrease in maximum sap flow for *Pinus contorta*, *Abies lasiocarpa*, *Populus tremuloides*, and
*Pinus flexilis* when soil moisture declined by 31.4%. Under the conditions of a saturated shallow
water table, forest transpiration could equal PET (Čermák and Prax, 2001). Brito et al. (2015)
found that the total canopy transpiration of *Pinus canariensis* increased by 133% in a wet year
than that in a normal year. Canopy transpiration variation showed significant correlation with
soil water content variation (Fig. 5). In addition to soil moisture, the low regression coefficient
can be attributed to the influence of various environmental factors (Bosch et al., 2014; Brito et al.,
2015; Chen et al., 2014c). Among these factors, VPD and solar radiation can trigger a timely
response in transpiration, while the influence of soil water is reflected over a longer temporal
scale (Chen et al., 2014a; Shen et al., 2015).
As Chen et al. (2014c) indicated that the sensitivity of stomatal response to drought stress can be
expressed by the frequency distribution of maxmimum sap flux density. The increased frequency
of maximum sap flux density earlier in the day (before 14:00 LT) suggested an enhanced
stomatal sensitivity to avoid high VPD (Fig. 7). Studies have shown that the effectiveness of
stomatal conductance induced by VPD fluctuation could result in the variation of transpiration
rate (Addington et al., 2004; Igarashi et al., 2015), and a decline in canopy conductance with
increasing VPD is an indicator of physiological restrictions to transpiration (Chang et al., 2014a;
Shen et al., 2015). This occurred to avoid any negative leaf water potential and xylem cavitation
(Addington et al., 2004; Wang et al., 2014b). When soil conditions are severely stressed or in a





prolonged period of VPD tension, it is inevitable that the varying degrees of embolisms can be
caused by runaway cavitation (Vergeynst et al., 2015), which could trigger a series consequences,
such as reducing water transport, and stomatal closure (Pataki et al., 2000). This would explain
why canopy conductance decreased logarithmically with VPD and reduced sharply when soil
water condition changed from wet to dry (Fig. 6). In time and space, soil moisture plays an
important role in connecting environmental fluctuations and vegetation transpiration (Brito et al.,
2015; Chen et al., 2014a). Similar to the conclusions drawn by Shen et al. (2015), our results
showed that canopy transpiration and canopy conductance of *P. tabulaeformis* were 6.9% and
3.9% higher at terrace site than that at slope site. Our results suggested that the impact of
terracing on transpiration could be explained by the response of canopy transpiration to other
environmental factors under different soil water conditions.
**4.3 Implications of this study**
Under water stress, species tend to adjust their water consumption to avoid reaching water
potential values that could produce irreversible damage (Chirino et al., 2011). Depending on
their drought avoidance mechanisms, species can be classified into water-spender or water-saver
types (Chirino et al., 2011). In this context, *P. tabulaeformis* showed lower sap flux density
under drier water conditions (Fig. 3) and reduced canopy conductance with an increasing VPD
(Fig. 6 b and d). Therefore, *P. tabulaeformis* can be classified as a water-saver species
(Heilmeier et al., 2002). Yang et al. (2008) indicated that in the semiarid Loess Plateau, *P.*
*tabulaeformis* uses water more efficiently than *Robinia pseudoacacia*, and *Malus pumila*. Similar
results were found in mixed forests of different ages (Chang et al., 2013) and different species
(Chen et al., 2014b; Nie et al., 2005). In dry regions, *P. tabulaeformis* might be a good
drought-resistance species that could help control soil loss and improve the ecological



251 environment. In this study, it was found that terracing significantly improved soil water

252 conditions. It captured 113.1% more REW than that at the slope site (Fig. 2), and it increased

253 canopy transpiration significantly (Fig. 4). Meanwhile, the average DBH, sapwood area, and

254 crown projected area were 12.0%, 18.8%, and 63.5% higher, respectively, at the terrace site than

255 that at the slope site, and the crown projected area showed statistical significance ($p < 0.05$)

256 (Table 1). Just as Wang et al. (2012a) have indicated, in drylands, the most efficient use of water

257 is to maximize the productive water loss ($T$) and minimize the unproductive water loss ($E$).

258 Terracing increases the accumulation of the limited water supply, making more water available

259 for transpiration and growth and thus, improving the efficiency of water use.

260 **5 Conclusions**

261 In this study, the soil water content variation and daily canopy transpiration of *Pinus*

262 *tabulaeformis* were studied over two consecutive growing seasons (2014–2015) in a typical

263 semiarid area of the Loess Plateau in China. The effects of terracing on soil water content,

264 canopy transpiration, and canopy conductance were investigated. Terracing was found to have a

265 statistically significant positive effect on soil water content. *P. tabulaeformis* in the terrace site

266 showed significantly higher canopy transpiration than that in the slope site ($p < 0.05$), and the

267 variation between the terrace and slope sites increased with soil water content variation ($p <$

268 $0.0001$, $R^2 = 0.20$). The impact of terracing on transpiration could be expressed through the

269 response of canopy transpiration to other environmental factors. Terracing increased the

270 accumulation of the limited water supply, providing a greater amount of water for transpiration

271 and growth. For sustainable vegetation restoration in semiarid regions, the adoption of terracing

272 could be a technique worthy of consideration.



**Acknowledgements**
This research was supported by the National Natural Science Foundation of China (41371123,
41390462) and the Innovation Project of the State Key Laboratory of Urban and Regional
Ecology of China (SKLURE2013-1-02).





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



## Tables

Table 1. Description of the study sites in Anjiapo catchment.

| | Parameter | Type | |
| --- | --- | --- | --- |
| | | Slope | Terrace |
| Geographical | Plot area (m$^2$) | 100 | 100 |
| parameters | Slope aspect | N | N |
| | Slope position | Middle | Middle |
| Biological | Dominant plant | Chinese pine | Chinese pine |
| parameters | Sample/total number | 6/14 | 6/21 |
| | DBH (cm) | 12.90[a] $\pm$ 3.66 | 14.45[a] $\pm$ 2.40 |
| | Sapwood area ($A_s$,cm$^2$) | 99.09[a] $\pm$ 44.87 | 117.74[a] $\pm$ 33.16 |
| | Crown projected area ($A_c$, m$^2$) | 9.55[a] $\pm$ 3.56 | 15.61[b] $\pm$ 5.11 |

*Note:* slope aspect and slope position were measured by compass; DBH is the diameter at breast height for trees, each parameter was measured from 2014 to 2015. Means without common letters are significantly different at $p < 0.05$ according to *t*-test.



1    **Figure Legends**

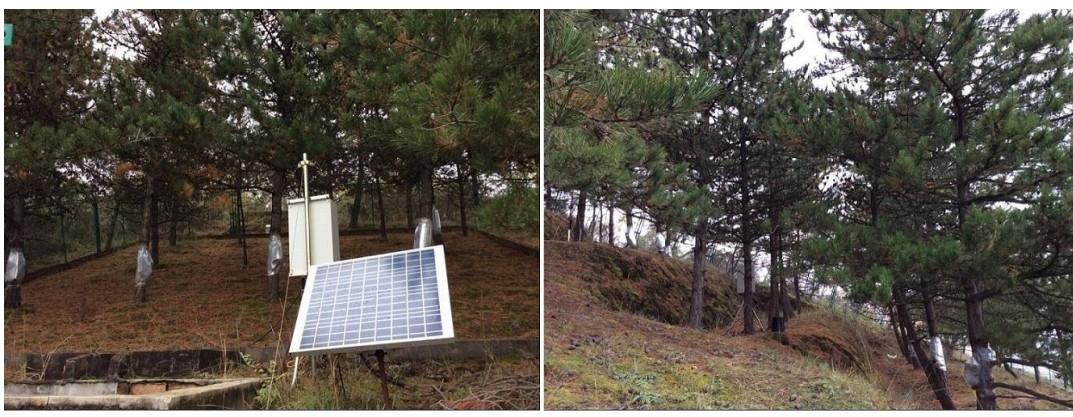

2                          Slope site                                              Terrace site

3    Fig. 1 Site photographs of the slope and terrace sites.





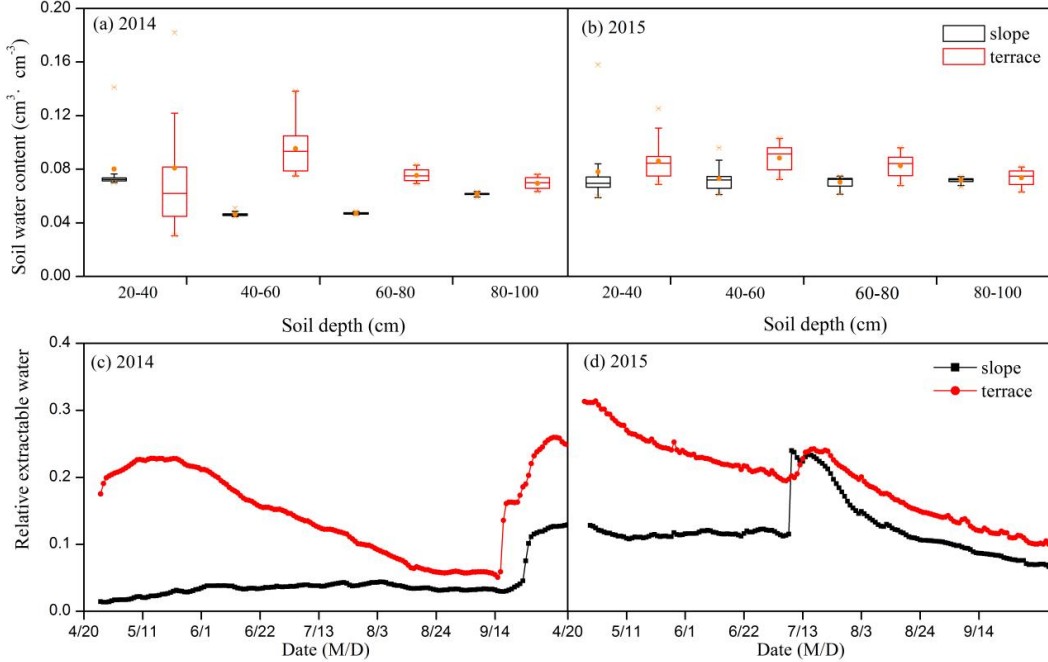

2 Fig. 2 Soil water content of different layers (a and b) and relative extractable water in the 20–100

3 cm layer (c and d) between the slope and terrace sites during two consecutive growing seasons

4 (2014–2015).





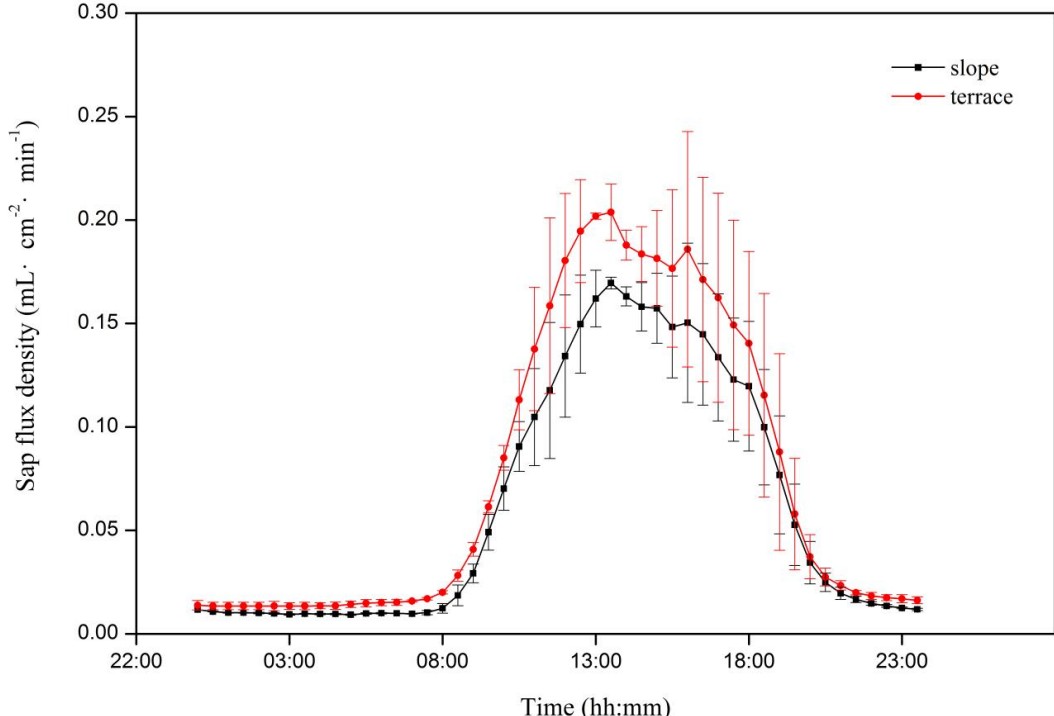

2    Fig. 3 Diurnal time courses of sap flux density at the slope and terrace sites. Data represent

3    means ± standard deviation (n = 3).



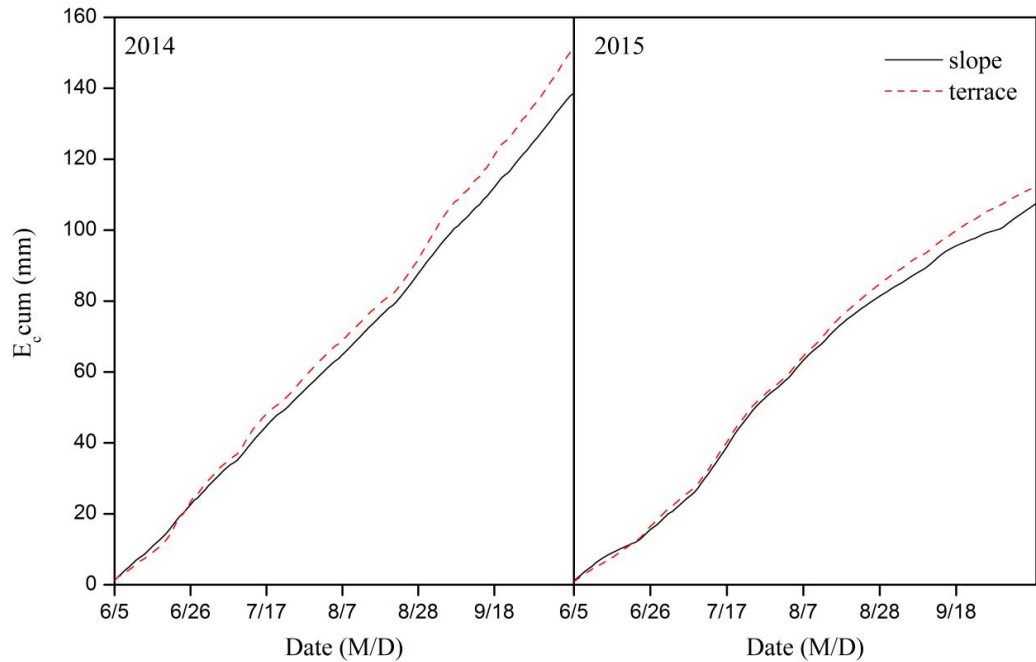

2    Fig. 4 Variation of accumulated canopy transpiration ($E_c$ cum) during two consecutive growing

3    seasons (2014–2015).





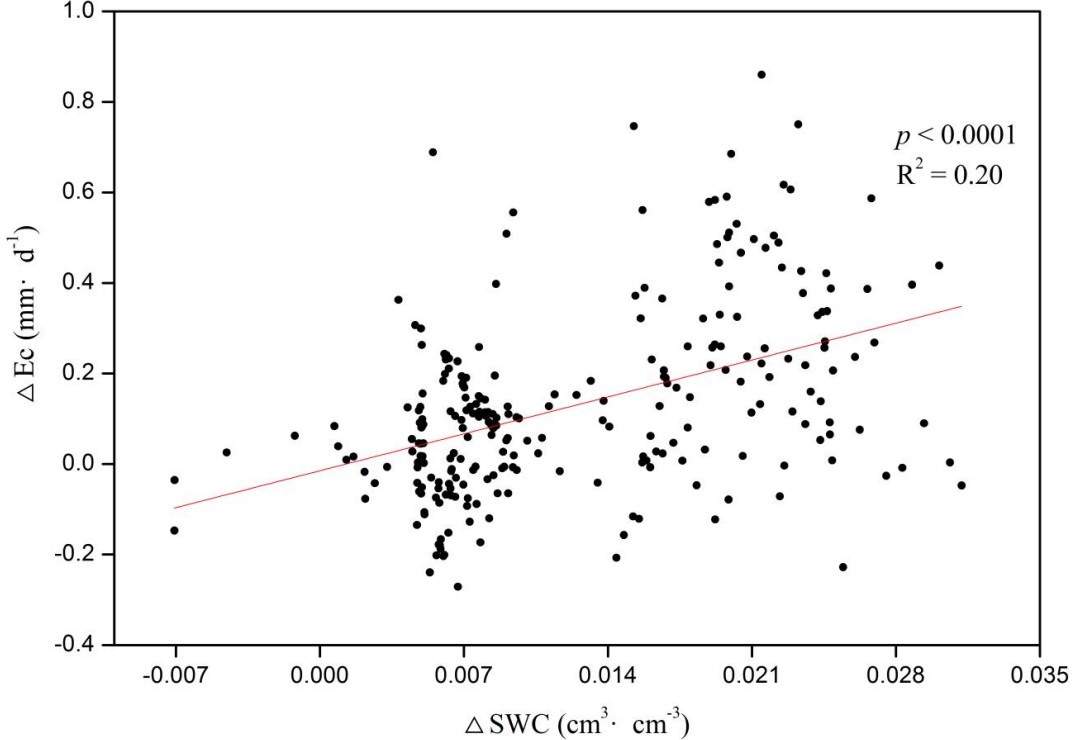

2 Fig. 5 Correlation of $E_c$ variation in response to SWC variation within 100 cm during two

3 consecutive growing seasons (2014–2015). Data of the slope site are the baselines subtracted by

4 those of the terrace site to assess the relationship between $E_c$ variation and SWC variation.



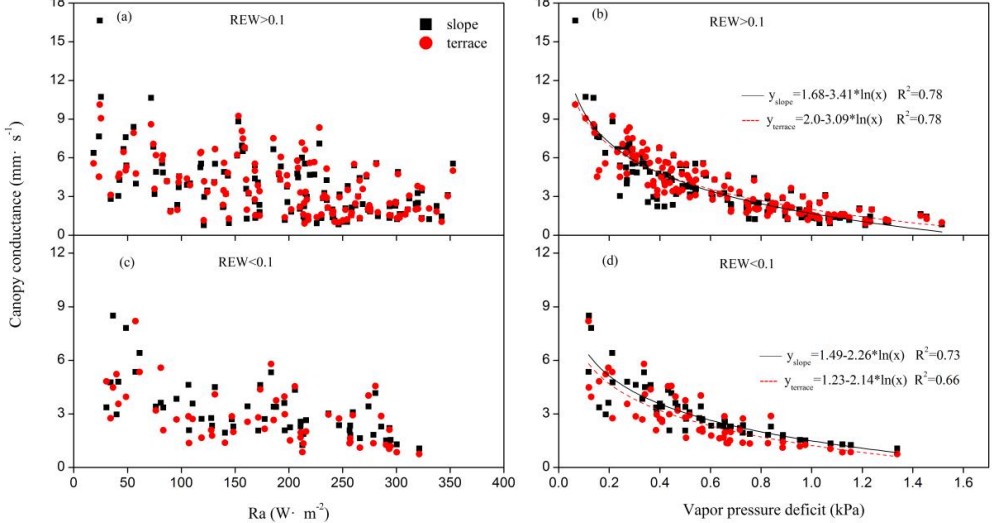

2    Fig. 6 Relationships between canopy conductance and solar radiation (a and c) versus

3    relationships between canopy conductance and vapor pressure deficit (b and d) under relatively

4    wet (REW > 0.1) and dry (REW < 0.1) soil conditions.


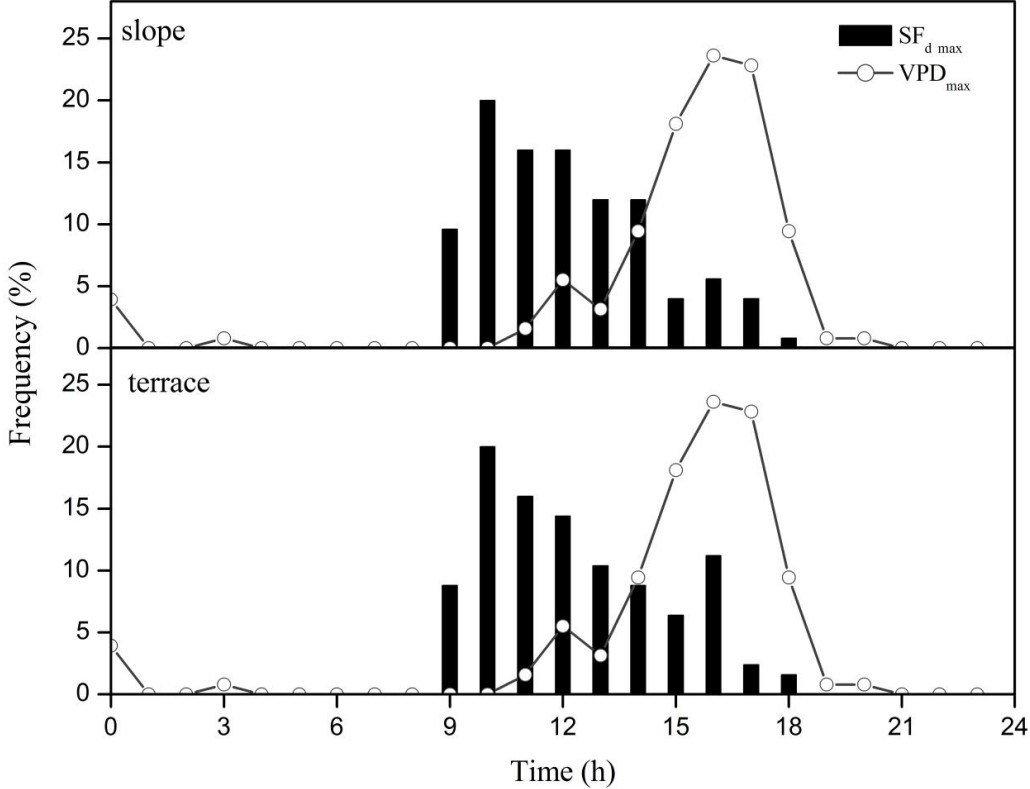

2    Fig. 7 Frequency distribution of maximum sap flux density ($SF_{d, max}$) and VPD peak times in the

3    form of diurnal patterns at contrasting sites for the comparison of $SF_d$ - VPD evolvement patterns.

4    Data sets cover the growing seasons in 2014 and 2015.