# Peer review of "Effects of terracing on soil water and canopy transpiration of Chinese pine plantation in the Loess Plateau, China"

_Hydrology and Earth System Sciences, 2016_

## Referee Comment (RC1) · Anonymous Referee #1 · 21 Jun 2016

The manuscript presents a study to explore and possibly quantify the effect that terracing might have on soil moisture and canopy transpiration of pine plantations. The main conclusion of the study is that terracing increases soil moisture, with a consequent enhancement of transpiration and growth; accordingly, terracing can be an effective way for re-vegetation in arid regions.

My major concern with the manuscript is that, although the conclusions of the study seem reasonable, the design of the experiment does not seem to really address the research question that the Authors want to answer. There are many reasons why soil moisture and transpiration might be different in the 2 plots selected for this study; since a lot of details on the experiment are not given, it is difficult to isolate the effect of slope

or terrace on the water balance.

I also believe that some of the technical parts (e.g., calculation of REW and results on $g_c$) are not fully correct and could be largely improved.

I listed below some specific comments and suggestions.

- Lines 24-25: re-phrase.

- L37-38: re-phrase.

- L80-81: this statement should be backed up with some additional information. What is the water table depth? Are there measurements of root depth? How can the Author be certain that trees are not reaching groundwater or the capillary fringe?

- L83: it would be good to say here how large the plots are (100 m$^2$ according to Table 1) and what percentage of the entire slope and terrace they cover.

- L85: when were the trees planted? Do they have the same age in both sites? What is the tree density and how many trees are in each plot?

- L94-96: how many soil moisture profiles were in each plot? It looks like there was only one profile per plot. Considering that the plots are about 10 m x 10 m, is one profile enough to compare the two plots? The Authors should justify this.

- If the two plots have the same soil and vegetation, REW should be the same. By using different $\theta_{max}$ and $\theta_{min}$ in the two years and in the two plots, the comparison between plots becomes rather confusing. I would calculate REW based on the minimum and maximum $\theta$ in the two years irrespective of the plot, so that the comparison between plots is done using the same scaling of soil moisture data.

- Eq. 4: how was the xylem (sapwood) area measured? Since the DBH of the trees in the two plots is rather different (that is why it is important to know whether the trees were planted at the same time), larger $A_s$ will be associated with larger transpiration rates. Errors in estimating $A_s$ are going to affect the estimates of $E_c$.

- Eq. 5: it is said that 6 trees per plot were instrumented with sap-flow sensors. Is the average of these fluxes used in this equation? It is not really clear how the measurements in 6 trees were used in Eqs. 4 and 5.

- L142-143: the Authors should explain in detail why the measurements at 10 cm were not used. Since it is likely that the majority of the root system of pines is in the first 30-40 cm of soil, using soil moisture data only at 30 cm and below will reduce the ability to link transpiration and soil moisture.

- L193: what does 'in each layer' refer to here?

- Table 1: what is the meaning of 6/14 and 6/21

- Figure 1 could be removed

- Fig 2: I would add rainfall in the figures with REW (and I would calculate REW as suggested above). It is said that the 2 plots are under the same conditions; however, looking at the delay between the increases in REW in the two plots, it seems that there might be differences in rainfall.

- Fig 3: in the text, it is said that there are 6 trees with sap flow, while here n=3. Why?

- Fig 5: it should be explained better what this figure shows. It looks like the two axes are $\theta_{terrace} - \theta_{slope}$ and $E_{terrace} - E_{slope}$. Is that correct? Are these daily values or values every 30 minutes? What is the message of this figure?

- Fig 6: I would show here how $E_c$ (not $g_c$) relates to solar radiation and VPD.

- Fig 7: this is not a frequency distribution of maximum sap flux density. It seems that this figure shows the percentage of time the daily maximum flux and the daily maximum VPD occurred at certain times of the day.

---

## Referee Comment (RC2) · Anonymous Referee #2 · 1 Jul 2016

This study quantifies the terracing on soil water dynamics and vegetation water use associated with tree plantation in the Loess plateau of China. The positive effects of terracing on conserving soil water and promoting canopy transpiration are highlighted comparing to natural slopes. Some useful and important first hand data and information are provided but somewhat site-specific. I think the authors may need to add more materials or at least some discussion on the large scale impact across a broader region. Probably a modeling work can be helpful to understand the terracing effect across large space and over longer time period. Some specific comments and suggestions are posted in the following text.

- Line 25: "considerable implications regarding forest management and water yields"

[Figure]

what are considerable implications? Not clear and need a rephrase.

- Line 68: "artificial plants" is this a right word?

- Line 95-96: how many soil moisture stations are employed in each stand? How close are they to the roots? It seems to me only one station is there for each stand, and if so, the soil moisture measurement would not be representative enough and could be completed biased if too close to or far away the root area.

- Line 113-114: there are grammatical errors in the sentences.

- Line 143: how did you do the significant test without site replicates? If you have, you may need more explanation to clarify.

- Line 158-159: how did you do the significant test for the canopy transpiration? Are there measurements for different trees in each stand?

- Line 164: proportion of rainfall? Need more explanation.

- Figure 5: What do you mean "soil water variation" and "canopy transpiration variation"? The figure description in text and caption are needed to be improved for understanding.

- Figure 6: need a new figure. The subpanels are not clear and axis labels are vague.

- Line 214: what does the "low regression coefficient" refer to? Must be related to some correlations but without clear description.

- Line 231: Canopy conductance decrease with VDP? Do you mean Increase of VDP? More clarification is needed throughout the whole manuscript on this problem.

- How about the weather difference between the two years in your study? Did the weather/climate difference have any impact on your study? In addition, the stands selected in the study are both located in north facing slopes, how about the south facing slopes? Qiao et al. (2015) and Zou et al. (2015) have used both experimental watershed data and modeling work to examine woody plant impacts on a broader region across climatic and physiographic settings in the southern Great Plains, USA. I think their studies could be useful references and provide some helpful hints to this study.

―――――――――――――――

---

## Author Comment (AC1) · 6 Jul 2016

Dear referees,

Thank you for your valuable comments concerning our manuscript. These constructive comments are very helpful for revising and improving our manuscript. We have studied the comments carefully and have made extensive changes. Revised portion are marked in red in the revised manuscript. The major changes in the manuscript and the responses to the referee's comments are listed as following:

Comment 1: Lines 24-25, re-phrase.

Response: Thank you for your detailed comments. According to the reviewer's comment, we have changed the sentence of "Transpiration as an important role part of the soil-plant-atmosphere continuum has considerable implications regarding forest management and water yields, ..." to "Transpiration constitutes an important part of the water budget in the soil-plant-atmosphere continuum. It can affect forest water yields and mechanism-based study on transpiration will provide theoretical guidance for forest management, ...".

Comment 2: L37-38, re-phrase.

Response: According to the reviewer's comment, we have changed the sentence of "Stomatal closure as an important physiological process was employed by plants to regulate water use and ..." to "Stomatal closure is an important physiological process employed by plants to regulate water use and ...". Furthermore, we had the manuscript polished professionally.

Comment 3: L80-81, this statement should be backed up with some additional information. What is the water table depth? Are there measurements of root depth? How can the author be certain that trees are not reaching groundwater or the capillary fringe?

Response: Thank you for your valuable advice. For this study area, previous studies have indicated that the soil depth varied from 40 to 60 m, deep percolation can be neglected and groundwater is unavailable for vegetation growth. We did not measure the root depth of pine, however, as mentioned in previous studies, the roots mainly distributed within 60 cm soil layers (Liu et al., 2007). We added references to support these statements.

Comment 4: L83, it would be good to say here how large the plots are (100 cm2 according to Table 1) and what percentage of the entire slope and terrace they cover.

Response: Thank you for the comments. As suggested, we added the plot size and coverage information.

Comment 5: L85, when were the trees planted? Do they have the same age in both

HESSD
sites? What is the tree density and how many trees are in each plot?

Response: Thank you for your valuable advice. By the end of the introduction we had mentioned that the pines were similarly aged, and now we added the specific year when these trees were planted and the mean canopy coverage of each plot. The tree density was also mentioned in Table 1.

Comment 6: L94-96, how many soil moisture profiles were in each plots? It looks like there was only one profile per plot. Considering that the plots are about 10 m  $\times$  10 m, is one profile enough to compare the two plots? The authors should justify this.

Response: Thank you for your valuable advice. For each plot we chose four profiles, with three profiles uniformly distributed in the upper, middle, and bottom of the plot and measured with TRIME-FM time domain reflectometry (TDR) twice a month, another one in the middle of the plot and measured with HOBO U30 continuously in every 10 minutes. We compared the data of each profile, and it showed that the soil water content monitored by U30 can well represent the plot. The related information has been added in Page 6, Line 95-100.

Comment 7: If the two plots have the same soil and vegetation, REW should be the same. By using different  $\theta$ max and  $\theta$ min in the two years and in the two plots, the comparison between plots becomes rather confusing. I would calculate REW based on the minimum and maximum  $\theta$  in the two years irrespective of the plot, so that the comparison between plots is done using the same scaling of soil moisture data.

Response: Thank you the comments and sorry about the confusion. For calculating the REW, we actually used the same  $\theta$ max and  $\theta$ min.

Comment 8: Eq. 4, how was the xylem (sapwood) area measured? Since the DBH of the trees in the two plots is rather different (that is why it is important to know whether the trees were planted at the same time), larger As will be associated with larger transpiration rates. Errors in estimating As are going to affect the estimates of Ec.
Response Thank you for the constructive comments. Sapwood area was estimated by differing the different colors between sapwood and heartwood. Furthermore, we also tested our results with previous studies and established an equation .

Comment 9: Eq. 5, it is said that 6 trees per plot were instrumented with sap flow sensors. Is the average of these fluxes used in this equation? It is not really clear how the measurements in 6 trees were used in Eqs. 4 and 5.

Response: Thank you for the comments. Data of individual tree were used to calculate the sap flux (Eq. 4) and canopy transpiration (Eq. 5). Then, when comparing the differences between slope plot and terrace plot, the average of sap flux and canopy transpiration were used.

Comment 10: L142-143, the authors should explain in detail why the measurements at 10 cm were not used. Since it is likely that the majority of the root system of pines is in the first 30-40 cm of soil, using soil moisture data only at 30 cm and below will reduce the ability to link transpiration and soil moisture.

Response: Thank you for your valuable advice. In 2015, the probe in 10cm was loose (apart from the soil), so the data were not correct. So in order to make the data consistent, when comparing the differences of soil water content between slope and terrace, the first layer (0~20 cm) were not calculated. We added the detailed reason in the manuscript.

Comment 11: L193, what does "in each layer" refer to here?

Response: Thank you for your comment. The words "in each layer" in the text means the different soil layers ( $20 \sim 40$  cm,  $40 \sim 60$  cm,  $60 \sim 80$  cm, and  $80 \sim 100$  cm). For each layer, terrace plot had higher soil water content than slope plot. We clarified this in the revision.

Comment 12: Table 1, what is the meaning of 6/14 and 6/21

Response: Thank you for your comment. In this table, the 6/14 and 6/21 means the
sample number/ total number of trees in slope plot and terrace plot, respectively.

Comment 13: Figure 1 could be removed

Response: As suggested, we moved it to supplementary materials.

Comment 14: Fig 2, I would add rainfall in the figures with REW (and I would calculate REW as suggested above). It is said that the 2 plots are under the same conditions; however, looking at the delay between the increases in REW in the two plots, it seems that there might be differences in rainfall.

Response: Thank you for your valuable advice. As suggested, we added the rainfall in the figure. The differences between the increases in REW in the two plots may be caused the different soil characteristics and the different percolation rate. Studies showed that land preparation may cause differences in soil characteristics.

Comment 15: Fig 3, in the text, it is said that there are 6 trees with sap flow, while here n=3. Why?

Response: Thank you for your detailed comments and sorry about the confusion. All of the 12 trees (six for each plot) were used to calculate the sap flux density, and the number (n=3) represents the days that used for comparing the differences of diurnal variation of sap flux density between two plots to eliminate the error caused in one individual day. We clarified this in the revision.

Comment 16: Fig 5, it should be explained better what this figure shows. It looks like the two axes are  $\theta$  terrace  $-\theta$  slope and E terrace – E slope. Is that correct? Are these daily values or values every 30 minutes? What is the message of this figure?

Response: Thank you for your detailed comments. It is correct that the two axes are  $\theta$  terrace –  $\theta$  slope (x axis) and E terrace – E slope (y axis). Daily values were used in this figure. Soil water content was almost stable in a short temporal scale. This figure was mainly used to indicate that the differences between slope plot and terrace plot can significantly affect the canopy transpiration. We clarified this in the revision.

**HESSD**
Comment 17: Fig 6, I would show here how Ec (not gc) relates to solar radiation and VPD.

Response: Thank you for your valuable advice. In our opinion, the correlation between gc and solar radiation and between gc and VPD can more directly reflect the plant response to drought.

Comment 18: Fig 7, this is not a frequency distribution of maximum sap flux density. It seems that this figure shows the percentage of time the daily maximum flux and the daily maximum VPD occurred at certain times of the day.

Response: Thank you for your detailed comments. We are sorry for the inaccurate expression. This figure shows the frequency distribution of the time for the maximum sap flux density occurred and the VPD peak times in the form of diurnal patterns. And the corresponding revision has been made in the manuscript.

Thank you again for your constructive comments and suggestions.

**HESSD**

---

## Author Comment (AC2) · 6 Jul 2016

Dear referees,

Thank you for your valuable comments concerning our manuscript. These constructive comments are very helpful for revising and improving our manuscript. We have studied the comments carefully and have made extensive changes. Revised portion are marked in red in the revised manuscript. The major changes in the manuscript and the responses to the referee's comments are listed as following:

Comment 1: Line 25, "considerable implications regarding forest management and water yields" what are considerable implications? Not clear and need a rephrase.

Response: Thank you for your comments. The sentence has been changed into "Transpiration constitutes an important part of the water budget in the soil-plant-atmosphere continuum. It can affect forest water yields and mechanism-based study on transpiration will provide theoretical guidance for forest management, . . .".

Comment 2: Line 68, "artificial plants" is this a right word?

Response: The words have been changed into "introduced plants".

Comment 3: Line 95-96, how many soil moisture stations are employed in each stand? How close are they to the roots? It seems to me only one station is there for each stand, and if so, the soil moisture measurement would not be representative enough and could be completed biased if too close to or far away the root area.

Response: Thank you for your detailed comments. For each plot, there is only one soil profile monitored automatically with the soil moisture station (U30). It is located in the middle of the plot about 50 cm away from the trunk. Meanwhile, there are another three soil profiles uniformly distributed in the upper, middle, and bottom of each plot, which were monitored by TRIME-FM time domain reflectometry (TDR) twice a month, to calculate the soil water content. By comparing the data of each profile, it showed that the soil water content monitored by U30 can well represent the plot.

Comment 4: Line 113-114, there are grammatical errors in the sentences.

Response: Thank you for your detailed comments. We have revised the sentence into " is the maximum value of with zero transpiration assumed".

Comment 5: Line 143, how did you do the significant test without site replicates? If you have, you may need more explanation to clarify.

Response: Thank you for your valuable advice. We have multiple monitoring locations (replicates) within each treatment. In the "2.4 Statistical analysis", we mentioned that since the data of soil water content were auto-correlated in the time series, non-parametric tests of significance were used. The Wilcoxon rank sum test, also known

as the Mann-Whitney U test, was used to test the differences in soil water content between two plots.

Comment 6: Line 158-159, how did you do the significant test for the canopy transpiration? Are there measurements for different trees in each stand?

Response: Thank you for your valuable advice. For each plot, six individual trees were monitored, and the canopy transpiration were calculated separately and used for statistical testing. When compared the differences between two plots, we used the average canopy transpiration.

Comment 7: Line 164, proportion of rainfall? Need more explanation.

Response: We are sorry for the unclear expression. And the sentence has been change into "The corresponding proportions of PET at the terrace site were 35.7

Comment 8: Figure 5, what do you mean "soil water variation" and "canopy transpiration variation"? The figure description in text and caption are needed to be improved for understanding.

Response: We are sorry for the unclear expression. The "soil water variation" and "canopy transpiration variation" represents $\theta$ terrace $-\theta$ slope and Ec terrace – Ec slope, respectively. The figure was mainly used to indicate that the differences between slope plot and terrace plot can significantly affect the canopy transpiration. The sentence in the text has been changed into "Furthermore, we examined that the increase of soil water content can make a significant increase in canopy transpiration." And the description of the figure has been changed into "Fig. 5 Correlation of Ec increment (Ec terrace – Ec slope) in response to SWC increment (SWC terrace – SWC slope) within 100 cm during two consecutive growing seasons (2014–2015)."

Comment 9: Figure 6, need a new figure. The subpanels are not clear and axis labels are vague.

Response: Thank you for your valuable advice. As suggested, we have made a new

figure.

Comment 10: Line 214, what does the "low regression coefficient" refer to? Must be related to some correlations but without clear description.

Response: Thank you for your valuable advice and we are sorry for the confusion. The "low regression coefficient" means the R2=0.20 in Figure 5. And the corresponding revision were made in the manuscript.

Comment 11: Line 231, canopy conductance decrease with VPD? Do you mean increase of VPD? More clarification is needed throughout the whole manuscript on this problem.

Response: Thank you for your valuable advice. The canopy conductance decrease with VPD means the canopy conductance decrease with the increase of VPD. As suggested, we have revised related the sentences throughout the whole manuscript.

Comment 12: How about the weather difference between the two years in your study? Did the weather/climate difference have any impact on your study? In addition, the stands selected in the study are both located in north facing slopes, how about the south facing slopes? Qiao et al. (2015) and Zou et al. (2015) have used both experimental watershed data and modeling work to examine woody plant impacts on a broader region across climatic and physiographic settings in the southern Great Plains, USA. I think their studies could be useful references and provide some helpful hints in this study.

Response: Thank you for your valuable advice and we have searched and read these two articles carefully. There was no significant difference of climatic indicators between 2014 and 2015. Although the canopy transpiration can be affected by climate, the two plots were next to each other, the influences caused by climate can be neglected. Similarly, the slope aspect influences on canopy transpiration were the same. It did not affect the difference between slope and terrace. So we think the evaluation of

the effects of terracing on soil water content and canopy transpiration was reasonable. Meanwhile, the articles were cited in the revision.

Thank you again for your valuable comments and suggestions.

[Figure]

Fig. 1.